# Description of an Ultrasound-Guided Erector Spinae Plane Block and Comparison to a Blind Proximal Paravertebral Nerve Block in Cows: A Cadaveric Study

**DOI:** 10.3390/ani12172191

**Published:** 2022-08-25

**Authors:** Olivia D’Anselme, Amanda Hartnack, Jose Suarez Sanchez Andrade, Christian Alfaro Rojas, Simone Katja Ringer, Paula de Carvalho Papa

**Affiliations:** 1Department of Clinical Diagnostics and Services, Section of Anaesthesiology, Vetsuisse Faculty, University of Zurich, 8057 Zurich, Switzerland; 2Department of Farm Animal, Vetsuisse Faculty, University of Zurich, 8057 Zurich, Switzerland; 3Department of Clinical Diagnostics and Services, Diagnostic Imaging Large Animals, Vetsuisse Faculty, University of Zurich, 8057 Zurich, Switzerland; 4Institute of Veterinary Anatomy, Vetsuisse Faculty, University of Zurich, 8057 Zurich, Switzerland

**Keywords:** erector spinae plane block, cattle, regional anaesthesia, spinal nerves, paravertebral anaesthesia

## Abstract

**Simple Summary:**

One of the most common surgical procedures performed in cattle is abdominal surgery, and this procedure is routinely performed in the standing adult animal. This procedure is performed using local anaesthetic techniques in order to eliminate pain and reduce the need for chemical and physical restraints. One of the most common locoregional anaesthetic techniques is the proximal paravertebral nerve block, in which a local anaesthetic is used to desensitize the nerves providing sensation to the flank region. The use of ultrasound-guided techniques has been demonstrated in multiple species, including humans, to be more accurate, safe and efficacious than traditional blind techniques. The aim of this paper was to develop an ultrasound-guided block that would target the nerves desensitized in the paravertebral nerve block of the flank. Additionally, our study aimed to evaluate the two techniques (blind and ultrasound-guided) in cow cadavers. The use of ultrasound guidance resulted in an apparent improvement in accuracy, and the blind technique remains a reliable approach. Further studies are warranted to develop and to evaluate the ultrasound-guided technique in live animals.

**Abstract:**

The proximal paravertebral nerve block is commonly used to provide anaesthesia to the flank during standing surgical procedures in adult cattle. It has been reported that additional anaesthetic infiltration may be necessary to provide complete anaesthesia. In humans as well as animal species, another technique—the ultrasound (US)-guided erector spinae plane block (ESPB)—has been described. The goal of the present study was to develop and investigate an US-guided ESPB in comparison to a blind proximal paravertebral nerve block (PPNB) in cow cadavers. In 10 cadaver specimens, injections of methylene blue-lidocaine (1:1) were performed at the level of T13, L1 and L2 vertebras, on one side doing an ESPB block and, on the other side, a PPNB. Five cadavers were injected with high (40 mL per injection for PPNB and 20 mL for ESPB) and five with low (20 and 15 mL, respectively) volumes of injectate. For the ESPB, the ultrasound probe was oriented craniocaudally, and the ventral-cranial aspect of the articular processes (T13, L1 and L2) was targeted for injection. The dye spreading was evaluated by dissection. The landmarks for US-guided injection were easily visualized; however, injections were accidentally performed at T12, T13 and L1. Nevertheless, L2 was stained in 60% of ESPBs. Epidural spreading was observed with both techniques and all volumes. Viscera puncture was reported in two PPNBs. The ESPB resulted in similar nerve staining compared to the PPNB while using a lower volume of injectate. Even better staining is expected with a T13-L2 instead of a T12-L1 ESPB approach. Further studies are warranted to evaluate the clinical efficacy.

## 1. Introduction

In cattle, pain management relies on a restrained legal choice of drugs and a balance between the economic constraints of the production enterprise and the need to mitigate discomfort [1]. One of the most common surgical procedures performed in cattle is abdominal surgery [2,3]. Abdominal procedures are routinely performed with the animal standing using locoregional anaesthesia [3]. Locoregional anaesthesia allows standing surgical procedures in cattle with minimal physical and chemical restraints [4]. To provide local anaesthesia and analgesia to the flank regions, many techniques have been described: infiltration line block and inverted L-block, paravertebral nerve block (proximal and distal), segmental subarachnoid and high caudal epidural anaesthesia [4,5,6,7].

DeRossi et al. [8] and Nuss et al. [9] reported clinically that the paravertebral block (targeting the last thoracic (T13), first and second lumbar (L1, L2) spinal nerves) often fails to completely anesthetize the flank. To obtain a complete anaesthesia of the flank, a combination with a local infiltration technique may be necessary to provide complete analgesia to the skin.

The use of ultrasound (US)-guided techniques has become increasingly popular in human medicine as well as veterinary medicine. US-guided techniques present several advantages [10,11], including visualisation of the targeted area (fascia plane space or nerves) and a need of smaller volumes of local anaesthetics. Additionally, US-guided techniques may be safer than blind techniques by avoiding the puncture of perineural structures such as vessels, pleura and neuraxis [12]. Compared with peripheral nerve stimulation, US-guided nerve blocks are faster to perform and more successful [12,13]. They also have a shorter onset time, result in a better-quality block and last longer versus the blind technique and electrical nerve stimulation [13].

To achieve the same results as the proximal paravertebral nerve block (PPNB), an US-guided block has been described and used in humans and other species [14,15,16,17,18]: the erector spinae plane block (ESPB) first described by Forero in 2016 [14]. The ESPB consists of an injection of local anaesthetic in a fascial plane between the erector spinae muscles (iliocostalis, longissimus and spinalis) [14] and the tip of the transverse vertebral process. In humans, the ESPB is indicated for the treatment of acute or chronic pain following thoracic, abdominal, hip and spinal surgeries [14,19,20]. This block has been described in dogs [15,17], pigs [16] and recently in horses [18]. To the authors’ knowledge, this US-guided block has never been described before in cattle.

The present study aimed to describe an approach for an US-guided ESPB in fresh cow cadavers. The second objective was to describe and compare the dye distribution and nerve involvement after a blind PPNB and an US-guided ESPB performed on fresh cow cadavers. We hypothesized that the dye distribution and nerve involvement would be more targeted and hence more precise using the ESPB technique than the blind PPNB.

## 2. Materials and Methods

### 2.1. Animals

A total of 12 mature cows presented to the farm animal hospital of the Zurich Vetsuisse Faculty (University of Zurich) were euthanized due to illnesses unrelated to the present study and showed no spinal abnormalities. The number of cadavers included in this study was based upon the availability at the time of the study. The twelve cows were 4 +/− 1.45 years (mean +/− SD) old, all females, weighing 601 +/− 43 kg and with a body score condition of 3/5. Four were Braunvieh (Swiss Brown), seven were Holstein, and one was Fleckvieh. From the twelve cows, two were used for the pilot study and ten for the main study.

### 2.2. Anatomical Characterization and Pilot Study

The dorsal thoraco-lumbar region of bovines is normally targeted by anaesthetic procedures aiming to provide analgesia for standing surgeries. This region comprises the transition of the thoracal region, whose caudal limit is the 13th rib, and the Fossa paralumbalis (flank), whose dorsal limit is the Processa transversa of the lumbar vertebrae, caudal limit is the Ala ossis ilii (wing of ilium) and cranial limit is T13 (Figure 1).

Dissection of the dorsal region provides an overview and the anatomical organization of the spinal nerves. The spinal nerves found in this region are T13 and L1–L6, which leave the Medulla spinalis through the Foramen intervertebrale placed caudally to the corresponding vertebra. The spinal nerves are formed by a sensitive dorsal and a motor ventral root, which unite before going through the Foramen intervertebrale to build the Truncus nervi spinalis (Figure 2).

On the dorsal root (Radix dorsalis) proximal to the mentioned union, there is a Ganglion spinale formed by the neuron bodies of sensitive fibres covered by a connective tissue capsule. The Truncus nervi spinalis divides into a Ramus dorsalis, which innervate the skin and epaxial muscles, a Ramus ventralis, responsible for the innervation of hypaxial and body wall muscles as well as the skin in this region, and a Ramus communicans carrying the sympathetical innervation for the viscera, blood vessels and sweat glands. Some Rami ventrales of this specific region receive special names: L1 is called N. iliohypogastricus, and L2 is N. ilioinguinalis, which innervate the body wall until the cranial limit of the tight, peritoneum, wall muscles and skin. L3 (+L4) is N. genitofemoralis, which innervates the mammary glands, the layers surrounding the testis, the Funiculus spermaticus, the preputium and the skin medial to the tight. Together with T13, L1 and L2 are specially targeted for abdominal surgical procedures on standing cows.

The pilot anatomic dissection was performed on two dairy cow cadavers at the dorsum (T12-L5, including the skin and muscles, as wide as the width of the transverse processes). Their dorsum was removed after the autopsy, still fresh (not frozen). On both sides, different approaches were performed with goal of targeting the Ganglion spinale of Radix dorsalis.

The first step was the placement of the spinal needles under US-guidance targeting four locations based on the anatomical description of cow’s thoracolumbar vertebral column: cranial and caudal to the articular processes of the T13-L2 vertebrae and cranial and caudal to transverse processes. Immediate dissection of the specimen with the needles in place followed. This allowed us to visualise the exact location of the needle, indicating that the US-guided approach was efficient 100% of the time. The second step consisted of injecting dye solution at those four locations trying out different volumes (5, 10, 15 and 20 mL).

The result of this pilot study (see results section) allowed the development of the ESPB (described below) used to compare the efficacy of the blind PPNB and US-guided ESPB.

### 2.3. Cadaveric Study

In 10 cadaver specimens placed in sternal recumbency, injections of methylene blue-lidocaine (1:1) (Lidocain 2% Streuli; Streuli Pharma AG, Uznach, Switzerland) were performed at the level of T13, L1 and L2 vertebras, on one side doing a ESPB block and, on the other side, a PPNB. The first five cadavers were injected with a high (40 mL per injection for PPNB, 20 mL for ESPB) and the following five cadavers with a low (20 and 15 mL, PPNB and ESPB, respectively) volume injectate. The injections were made after euthanasia prior to necropsy.

#### 2.3.1. Ultrasound-Guided Technique: ESPB

An US scanner (GE Logid S8 Vet, XDClear, GE Healthcare, Zürich, Switzerland) with a macroconvex (1-6 MHz) was used to visualize the 13th rib as an anatomical landmark for the identification of thoracic vertebra (T)13. The last rib was identified as a convex in shape hyperechoic interface by placing the probe in a perpendicular angle to the vertebral column. This convex shape allowed differentiation from the transverse process of L1, since the transverse process appears in US as a straight and horizontally orientated hyperechoic interface. 

Once the last rib was identified in the US, the transducer was positioned parasagittally and orientated in long axis (Figure 3). The articular process of T13 was identified abaxial to the dorsal midline. Afterwards a 20 gauge, 3 ½’’ (0.9 × 90 mm) spinal needle K-3 lancet point (Dahlhausen & Co. GmbH, Köln, Germany) was advanced in-plane with the US transducer and oriented craniocaudally towards the cranial aspect of the articular processes of T13. Needle advancement stopped when the needle was visibly disappearing ventral to the cranial aspect of the articular process. 

When proper needle position was achieved, the stylet was removed, and a 20 mL syringe was attached. The solution was injected (one single injection) at the site where the spinal nerve was expected to exit the intervertebral foramen. The process was repeated for L1 and L2 always taking, as reference, the visualisation of the articular process of the vertebra L1 and L2, respectively (Figure 4).

#### 2.3.2. Blind Technique

All injections using blind PPNB technique were performed by an experienced surgeon (A.H) using the technique described by Anderson et al. [1].

### 2.4. Dissection

After the cows were euthanized and injected with the dye solution on both antimeres (in one using the blind technique and in the other the US-guided technique), a trained anatomist (C.A.R) dissected both sides, not knowing in advance which antimere was used for each of the two applied technics. Dissections were performed after autopsy, which took place maximally 24 h after euthanasia. All cadavers were kept below 4 °C before and after dissection.

The limits of dissection were equal for both sides: the Processus spinosus of the 12th thoracal vertebra as the dorso-cranial limit, Processus spinosus of the third lumbar vertebra as dorso-caudal limit and an imaginary line traced perpendicularly between the above cited Proccessus spinosi at 5 cm ventral to the Proccessus transversi as the ventral limit. The skin and subcutis were dissected and removed, as well as the Musculus (M.) cutaneus trunci. Caudally to the 12th and 13th Costae, Musculi (Mm.) retractor costae were also removed. The fascia toracolumbalis was opened and removed in the dorsal lumbar region, and then the M. iliocostalis lumborum bound to the M. longissimus lumborum were carefully dissected and separated from the medially placed Mm. multifidi.

The dorsal roots of spinal nerves T13-L2 were visualized, and the spreading of the dye along the nerves as well as the vertebral column could be measured. The anatomical specimens were turned to the ventral side where connective and adipose tissues, as well the M. psoas major were removed to visualize the ventral roots of T13-L2. The same measurements were performed for the ventral aspect. To finalize the observation of dye spreading, a median section of the vertebral column along the Processus spinosi was conducted using an electric saw (K440H, Kolbe GmbH—Foodtech, Elchingen, Germany). The medulla spinalis from T13-L4 was observed, and the spread of the dye in the Canalis vertebralis, dorsal roots, their Ganglion spinale and Dura mater was recorded for the left and right antimeres. Additionally, the Ligamentum lumbocostalis at T13 and the Ligamentum intertransversarium at L1 and L2 were also observed for dye spreading.

The structures were defined as stained if at least 2 cm of staining were measured.

## 3. Results

### 3.1. Anatomical Characterization and Pilot Study

When injecting cranially and caudally to the transverse processes of T13, L1 and L2, no staining of the nerves was observed. Injecting the dye solution at the caudal region of the articular processes of the vertebra resulted in staining of the joint but no staining of the nerves.

Finally, when injecting cranially to the articular processes, as described in Kramer et al. [21]), staining of the targeted area was achieved with 5 mL for the muscles and 15 mL for the nerves. From these results, we decided to choose the cranial region of T13–L1–L2 articular processes as landmarks to perform the US-guided ESPB using at least 15 mL of dye solution.

### 3.2. Cadaveric Study

#### 3.2.1. PPNB Blind Technique

Staining of the structures of the spinal nerves is described in Table 1, and an example is shown in Figure 5. Staining of the fascia, muscles, ligaments and bone structures is presented in Table 2. Epidural spread of the dye solution (Table 1 and Figure 6) was found in 8/15 (53%) for the high volume and 7/15 (47%) for the low-volume group. In two cadavers, viscera punctures (intra-renal/intra-spleen) and intra-abdominal staining of blue methylene were found. 

#### 3.2.2. ESPB Technique

In all animals, the anatomical landmark (articular process) for needle placement was identified with the ultrasound (US); however, injections were accidentally performed at T12, T13 and L1. Staining of the structures of the spinal nerves is described in Table 1, and an example is shown in Figure 5. Staining of the fascia, muscles, ligaments and bone structures is presented in Table 2. Epidural spread of the dye solution was found 5/15 (33%) for the high-volume group and 10/15 (67%) for the low-volume group. No viscera puncture or intra-abdominal spreading of the solution were found for the ESPB technique.

## 4. Discussion

To the authors’ knowledge, this is the first study describing an US-guided ESPB in cows. Although, the achieved approach was T12-L1 instead of T13-L2, nerve staining appeared to be at least as good as with the blind PPNB technique even with a lower injectate volume.

Locoregional anaesthesia is commonly used for standing laparotomy in cattle; however, US-guided techniques are not routinely applied. Only one study developed an US approach for a PPNB in cows [21] and one for a medial paravertebral nerve block in calves [22].

In our study, the development of the US ESPB was based on the dissection of the targeted anatomical region. The anatomic landmarks used are similar to those reported in previous equine studies [18,23,24]. Delgado et al. described a craniodorsal approach at the level of the transverse processes between two vertebrae [18]. However, in our pilot study, when injecting cranially and caudally to the transverse processes of T13, L1 and L2, no staining of the nerves was observed. 

The Processus articularis cranialis of the last thoracal vertebra and the first two lumbar vertebrae are much more prominent and cranially oriented in cows than in horses, which would explain why it was easy to identify as a landmark in our study but not used in Delgado et al. The injection of the dye solution at the caudal region of the cranial articular processes of the vertebra resulted in staining of the joint but no staining of the nerves. This could be explained by the shape of the articular process of the vertebrae in cows as described above. Due to this shape, staining of the nerves was easiest to achieve using a cranial approach to the Processus articularis cranialis.

Anatomical dissection of the region and the staining observed confirmed the efficacy of an ESPB technique in the present study. The approach was the same as previously described by Kramer et al. in 2014, who described the technique as an US-guided proximal paravertebral anaesthesia in cows [21]. The term ESPB was only introduced in human medicine in 2016 [14]. However, looking retrospectively, it was actually Kramer et al. who developed the first ESPB approach in cows. 

However, in their study, they used only 1 mL of dye solution per point of injection and concluded that it was difficult to stain the desired area. Since then, no other studies in adult cows were made. With our approach, we performed a similar technique to Kramer et al.; however, using a higher volume, and based on our results, the ESPB seems to be a promising technique when appropriate volumes are used.

While performing the dissection, we verified that the US technique injections were accidently performed one intervertebral space too cranial (T12-L1 instead of T13-L2). This observation was noticed during the dissection. However, L2 was stained with the high and low volume (table). This can be explained by the spreading of solution in the facial plane that allowed the staining of nerves up to L2. 

However, with a 15 mL volume, the nerve L2 was stained only on 2/5 for ganglion and rami communicantes and 3/5 for the dorsal and ventral branch. We would expect that, when the injections are performed at the location T13-L2, L2 would be stained in a higher percentage, as 100% staining of the targeted structures was achieved for T13 and L1 using a low-volume ESPB block.

It seems that 15 mL is a sufficient volume for an US-guided ESPB as better staining was achieved compared to the high-volume group. The rather superior staining (especially for the Ganglion spinale and Rami communicantes) observed using a low volume could be explained by a learning curve (Table 1). The researchers started the cadaveric study with the high-volume US technique group. In humans, it was reported that inexperienced ultrasound users can improve their hand-eye coordination within five subsequent trials in a simple model of a peripheral nerve block [25]. 

In another human study, anaesthesiology residents with little or no ultrasound experience can rapidly learn, as well as improve their speed and accuracy, in performing six trials of a simulated interventional ultrasound procedure [26]. Randomization of the treatment groups (low and high) volume would have prevented the influence of the learning curve on the present results.

Another finding from this study is the epidural spreading in both techniques. The epidural spreading has been described in previous ESPB cadaveric studies [18,27,28]. This is seen as a possible risk/complication for clinical application (for example collapse or loss of motor function during standing surgery). However, with our study, we were able to show through dissection that, also with the blind technique, epidural spreading does occur in 60% of injections with the clinically used volume of 40 mL. Although ataxia is commonly listed as a potential complication of PPNB, no case report has been found where the blind technique resulted in loss of motor function, ataxia or collapse. This would imply that also the epidural spreading observed with the ESPB would not be a concern during its clinical application. Additionally, DeRossi et al. (2010) reported the use of segmental dorsolumbar epidural anaesthesia in standing cattle with minimal ataxia [8].

The present study describes an US-guided ESPB in cows that may be used for local anaesthesia in standing laparotomy. As this block is US-guided, it offers the advantage of being more accurate and therefore more efficacious than traditional blind injections [13]. It is reported that US-guided injections allow the clinician to identify the desired landmarks as well as visualize the needle and the surrounding structures in real time. Ultimately, this ensures that the solution can be precisely injected at the intended site. US-guided blocks in humans are performed more rapidly and with superior accuracy to those performed with anatomical landmarks. 

This results in improvement of block quality, increased success rate, faster onset and reduced amount of local anaesthetic required [12]. In accordance, in the present study, lower volumes with the US technique led to equal staining as higher volumes using a blind technique. In humans, the use of US guidance improves block success and consequently postoperative analgesia reducing adverse effects of systemic analgesia, both of which are associated with faster discharge times and thus distinct economic savings [29].

The ESPB has the potential to be applicable in clinical practice as many large animal practices already use ultrasound regularly for other reasons, including reproductive, abdominal and thoracic examination. The landmarks are relatively easy to identify, and practitioners experienced using the ultrasound technique, coupled with their experience performing PPNBs, would be able to learn the technique quickly. Advantages in clinical practice would include increased accuracy and quicker onset time as well as lower volume requirements (lower risk of drug toxicity and lower costs). 

A higher block success would lead to reduced requirements of additional local or systemic analgesics during and after the procedure [30]. This would potentially also decrease the total surgical time as well as surgical complications caused by excessive movement during the procedure. Additionally, the current study demonstrated that intra-abdominal organ penetration is a potential complication of the blind and less with the US-guided technique. In two instances using the blind technique, the pathology department, responsible for the necropsies, reported intra-renal/intra-abdominal/intra-spleen infiltration of blue methylene. 

However, this is likely primarily a problem in recumbent animals (such as the cadavers used in our study), as the organs have less abdominal space, and the pressure against wall muscles increases, decreasing the thickness of these muscle layers, which results in increased chances of intra-abdominal organ puncture. While this risk is most likely decreased in the standing animal, it is not eliminated. Using the US technique, no intra-abdominal structures were inadvertently punctured in the current study.

As this is a descriptive study, we cannot draw conclusions using statistical analyses. However, with the US technique (even using a lower volume) a higher percentage of staining of the Rami ventrales, communicantes and the ganglion itself was observed. Anaesthesia of these structures would improve the quality of the regional anaesthesia and win in visceral analgesia in comparison with the blind techniques. The limitations of the present study are that it is a cadaveric study performed on a small number of specimens. 

The effect of the learning curve could not be ruled out as all the low-volume injections were performed after the high volume ones without randomization. However, the ESP block is an US-guided technique that appears promising for loco-regional anaesthesia for standing laparotomy in cows. Further studies are needed using the lower volume at the right spot and to test clinical efficacy of the technique.

## 5. Conclusions

This study showed that the US-guided ESPB technique resulted in at least a similar quality of nerve staining compared to the blind technique and using a lower volume of injectate. Further studies are necessary to validate the clinical application of the ESP block as a locoregional anaesthesia technique for standing laparotomy in cows.

## Figures and Tables

**Figure 1 animals-12-02191-f001:**
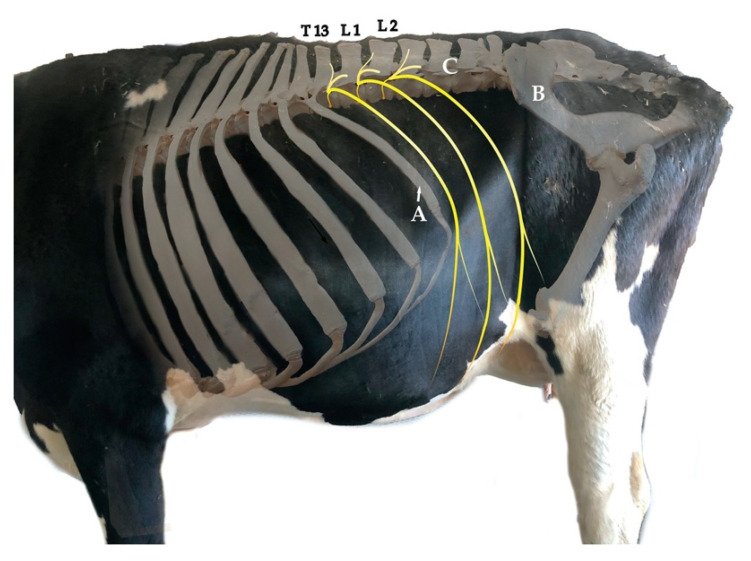
Superposed photographs from the left lateral view showing part of the skeleton, the trunk and limbs. The targeted spinal nerves for the proximal paravertebral nerve block are T13, L1 and L2, which are drawn in yellow. The flank is delimited cranially by the 13th rib (A), caudally by the Ala ossis ilii (B) and dorsally by the transverse processes of the lumbar vertebrae (C). Credits to C.A.R.

**Figure 2 animals-12-02191-f002:**
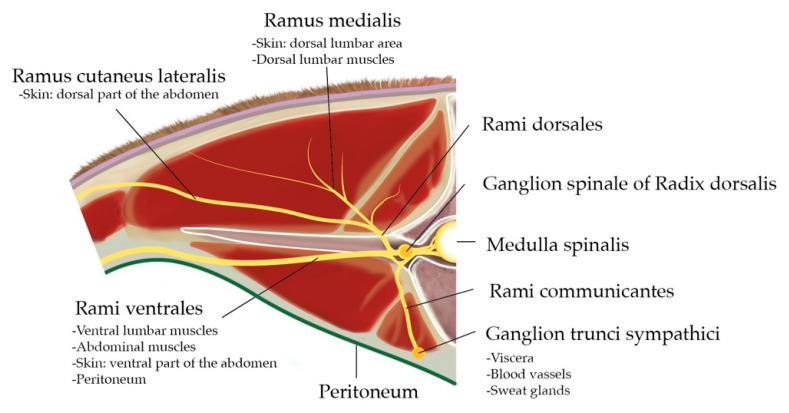
Cranial view at the intervertebral junction L1–L2: anatomical organisation of the spinal nerve L1 and area of innervation of the different Rami. Credits to C.A.R.

**Figure 3 animals-12-02191-f003:**
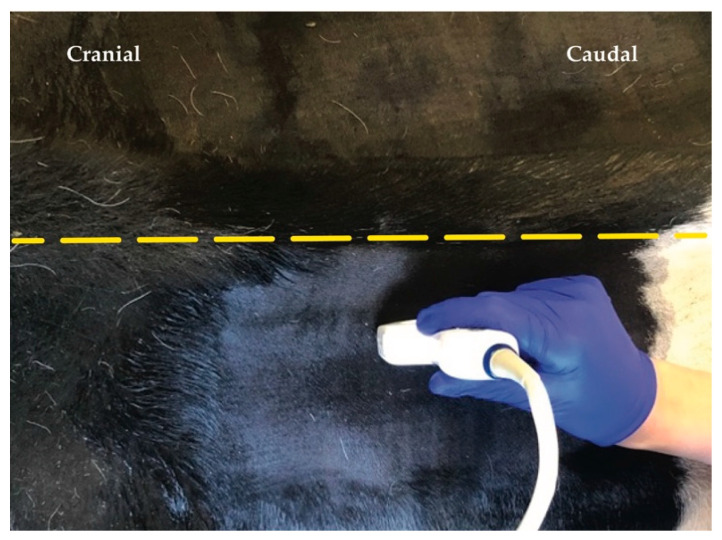
Position of the ultrasound probe parasagittally from dorsal midline (yellow dashed line).

**Figure 4 animals-12-02191-f004:**
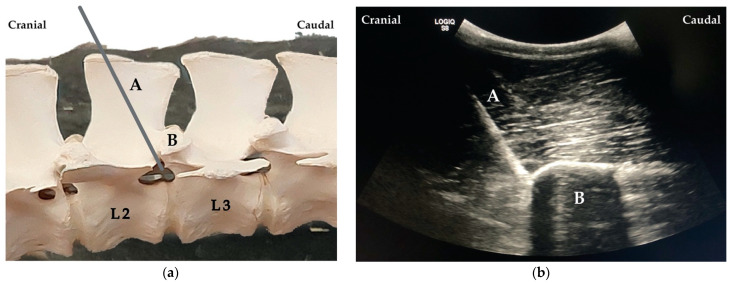
(**a**) Needle (A) placement cranial to the articular process of the second lumbar (L2) vertebra (B). (**b**) Ultrasound image showing the needle (A) cranially to the articular process of the vertebra L2 (B).

**Figure 5 animals-12-02191-f005:**
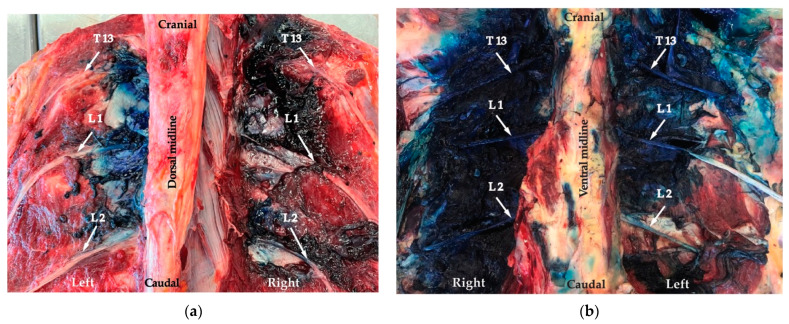
Dissection left side: the ultrasound-guided erector spinae plane block (ESPB) technique and right side: the proximal paravertebral nerve block (PPNB) blind technique. (**a**) Dorsal view: spread of the dye along the nerves (arrows). (**b**) Ventral view: spread of dye along the nerves (arrows).

**Figure 6 animals-12-02191-f006:**
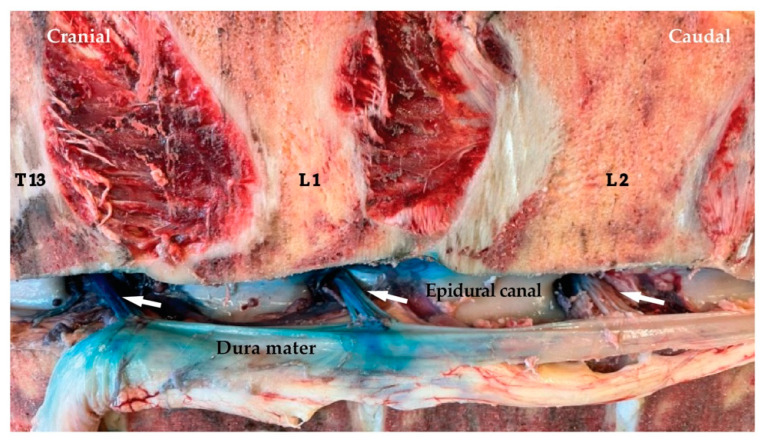
Staining of the dorsal roots of spinal nerves from T13 to L2 (arrows) on the right side (from blind proximal paravertebral nerve block (PPNB) low-volume (20 mL) group) after displacement of the Medulla spinalis. Note the staining of the Dura mater.

**Table 1 animals-12-02191-t001:** Structures of the spinal nerves stained in the different groups (blind proximal paravertebral nerve block (PPNB) high volume (40 mL), PPNB low volume (20 mL), ultrasound guided erector spinae plane block (ESPB) high volume (20 mL) and ESPB low volume (15 mL). Staining was considered positive when at least 2 cm were stained with methylene blue solution.

Truncus Nervi Spinalis Stained	Ganglion Spinale of Radix Dorsalis			Rami Dorsales			Rami Ventrales			Rami Communicantes			Epidural Spread			Total (*n*)
	T13	L1	L2	T13	L1	L2	T13	L1	L2	T13	L1	L2	T13	L1	L2	
PPNB high volume (40 mL)	2 (40%)	3 (60%)	4 (80%)	5 (100%)	5 (100%)	5 (100%)	4 (80%)	5 (100%)	4 (80%)	1 (20%)	3 (60%)	4 (80%)	3 (60%)	3 (60%)	2 (40%)	5
PPNB low volume (20 mL)	2 (40%)	3 (60%)	4 (80%)	5 (100%)	5 (100%)	5 (100%)	3 (60%)	3 (60%)	3 (60%)	2 (40%)	2 (40%)	3 (60%)	1 (20%)	3 (60%)	1 (20%)	5
ESPB high volume (20 mL)	3 (60%)	4 (80%)	2 (40%)	5 (100%)	5 (100%)	3 (60%)	5 (100%)	5 (100%)	3 (60%)	4 (80%)	4 (80%)	3 (60%)	3 (60%)	2 (40%)	0 (0%)	5
ESPB low volume (15 mL)	5 (100%)	5 (100%)	2 (40%)	5 (100%)	5 (100%)	3 (60%)	5 (100%)	5 (100%)	3 (60%)	5 (100%)	5 (100%)	2 (40%)	4 (80%)	4 (80%)	2 (40%)	5

**Table 2 animals-12-02191-t002:** Muscles and fascia stained in the different groups (blind proximal paravertebral nerve block (PPNB) high volume (40 mL), PPNB low volume (20 mL), ultrasound guided erector spinae plane block (ESPB) high volume (20 mL) and ESPB low volume (15 mL). Staining was considered positive when at least 2 cm were stained with methylene blue solution.

Anatomical Structures Stained	Fascia Thoracolumbalis			Longissimus Lumborum			Psoas Major			Processus Transversus			Ligamentum Lumbocostale T13/ Ligamenta Intertransversari L1 and L2			Total (*n*)
	T13	L1	L2	T13	L1	L2	T13	L1	L2	T13	L1	L2	T13	L1	L2	
PPNB high volume (40 mL)	3 (60%)	3 (60%)	2 (40%)	5 (100%)	5 (100%)	5 (100%)	5 (100%)	5 (100%)	5 (100%)	4 (80%)	4 (80%)	4 (80%)	5 (100%)	5 (100%)	5 (100%)	5
PPNB low volume (20 mL)	3 (60%)	4 (80%)	5 (100%)	5 (100%)	5 (100%)	5 (100%)	5 (100%)	5 (100%)	5 (100%)	2 (40%)	4 (80%)	4 (80%)	5 (100%)	4 (80%)	4 (80%)	5
ESPB high volume (20 mL)	2 (40%)	1 (20%)	2 (40%)	4 (80%)	4 (80%)	3 (60%)	5 (100%)	5 (100%)	5 (100%)	4 (80%)	4 (80%)	4 (80%)	5 (100%)	4 (80%)	4 (80%)	5
ESPB low volume (15 mL)	1 (20%)	1 (20%)	1 (20%)	5 (100%)	5 (100%)	4 (80%)	5 (100%)	5 (100%)	5 (100%)	5 (100%)	4 (80%)	4 (80%)	5 (100%)	4 (80%)	4 (80%)	5

## Data Availability

Not applicable.

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
