# Peer review of "Description of an Ultrasound-Guided Erector Spinae Plane Block and Comparison to a Blind Proximal Paravertebral Nerve Block in Cows: A Cadaveric Study"

_animals, 2022, doi:10.3390/ani12172191_

Round 1
Reviewer 1 Report
Dear Aunthors
This is a very well written paper. Having worked wth ruminants I understand all the traditional difficulties in perfomring proximal blocks in cattle for abdominal surgery.
This paper clearly desribes the 2 different techniques. Your illustrations are great.
However I found no mention of the breed of cattle or their weight (of the cadavers) or body condition score. Is to do with the ease of locating the landmarks,
I am assuming that these blocks were carried out with the animals recument in lateral position? There is a sentence (line 364) which mentions this...it might be better if it were mentioned earlier also to describe ow were the animals positioned for the blocks.
All in all a great paper! Well done!
Reviewer 2 Report
The manuscript “Description of an ultrasound-guided erector spinae plane block and the blind technique for proximal paravertebral nerve in cows: a cadaveric study” develop and investigate an ultrasound (US)-guided erector spinae plane block (ESPB) in comparison to a blind proximal paravertebral nerve block (PPNB) in cow cadavers. The subject is very interesting and it provides useful information for practical point of view, in order to eliminate pain and reduce the need for chemical and physical restraint. The experimental design is well done, but the material and method and result section are not well described. In my opinion, the manuscript is not suitable for publication in the present form.
Major revision
- Please, provide to rewrite the section of material and methods and results, putting the information in the right section. For example, move the sentence at lines 113-129 into introduction section; lines 207-209: move to material and methods.
- Line 144-145: why did You use these volume?
- Please, describe better the pilot study in material and methods.
- Lines 233-244 and Lines 254-262: put into other tables the information about the staining of muscles, fascia…
Minor revision:
- Lines 132-133: repeated sentence
- Review the reference section
Round 2
Reviewer 2 Report
The authors responded to my comments and suggestions clearly and comprehensively. In my opinion, the paper can be accepted.